# Effect of *Rubus idaeus* Extracts in Murine Chondrocytes and Explants

**DOI:** 10.3390/biom11020245

**Published:** 2021-02-09

**Authors:** Morgane Bourmaud, Mylene Zarka, Romain Le Cozannet, Pascale Fança-Berthon, Eric Hay, Martine Cohen-Solal

**Affiliations:** 1BIOSCAR Inserm U1132, Department of Rheumatology, Université de Paris, Hôpital Lariboisière, F-75010 Paris, France; morgane.bourmaud@inserm.fr (M.B.); mylene.zarka@inserm.fr (M.Z.); eric.hay@inserm.fr (E.H.); 2Naturex, Part of Givaudan SA, 250 rue Pierre Bayle, 84000 Avignon, France; romain.le_cozannet@givaudan.com (R.L.C.); pascale.fanca-berthon@givaudan.com (P.F.-B.)

**Keywords:** *Rubus idaeus*, chondrocytes, macrophages, osteoarthritis, inflammation

## Abstract

Osteoarthritis is characterized by cartilage loss resulting from the activation of chondrocytes associated with a synovial inflammation. Activated chondrocytes promote an increased secretion of matrix proteases and proinflammatory cytokines leading to cartilage breakdown. Since natural products possess anti-inflammatory properties, we investigated the direct effect of *Rubus idaeus* extracts (RIE) in chondrocyte metabolism and cartilage loss. The effect of RIE in chondrocyte metabolism was analyzed in murine primary chondrocytes and cartilage explants. We also assessed the contribution of RIE in an inflammation environment by culturing mice primary chondrocytes with the supernatant of Raw 264.7 macrophage-like cells primed with RIE. In primary chondrocytes, RIE diminished chondrocyte hypertrophy (*Col10*), while increasing the expression of catabolic genes (*Mmp-3*, *Mmp-13)* and reducing anabolic genes (*Col2a1*, *Acan*). In cartilage explants, *Rubus idaeus* prevented the loss of proteoglycan (14.84 ± 3.07% loss of proteoglycans with IL1 alone vs. 3.03 ± 1.86% with IL1 and 100 µg/mL of RIE), as well as the NITEGE neoepitope expression. RIE alone reduced the expression of *Il1* and *Il6* in macrophages, without changes in *Tnf* and *Cox2* expression. The secretome of macrophages pre-treated with RIE and transferred to chondrocytes decreases the gene and protein expression of *Mmp-3* and *Cox2*. In conclusion, these data suggest that RIE may protect from chondrocyte catabolism and cartilage loss in inflammatory conditions. Further evaluations are need before considering RIE as a candidate for the treatment for osteoarthritis.

## 1. Introduction

Osteoarthritis (OA) is the most common cause of arthritis affecting millions of people worldwide. OA affects 30% of people over 65 years and is one of the most expensive chronic diseases in developed countries. Because of its function-impairing nature, the burden is high in terms of prevalence and economic impact [1,2]. Pain and dysmobility are major drivers of clinical decision making and health service use. In contrast to this high prevalence, there is no treatment available to slow or treat the loss of cartilage. The pathophysiology is underpinned by several factors that combine mechanical overload, genetic, and hormonal factors. Several clinical risk factors are identified such as age, obesity, and trauma, whilst others are not modifiable. The breakdown of the cartilage matrix that characterizes OA generates pain and functional disability, a source of high morbidity. Chondrocytes, the unique specialized cells in the cartilage, ensure the renewal of the extracellular matrix which keeps the metabolic properties of the cartilage and its mechanical skills. In OA, chondrocytes form clusters and their activation results in increased secretion of matrix proteases (MMP-3, MMP-13, ADAMTS-5) capable of degrading the aggrecan and cleaving type II collagen network. Cartilage loss is also associated with a reduction in chondrocyte maintenance, a process that drives the commitment of chondrocytes into a hypertrophic maturation close to that observed during endochondral ossification. In addition to altered chondrocyte function, a synovial inflammation participates in cartilage loss through the invasion of immune cell infiltration that produces inflammatory cytokines that further activate chondrocyte catabolism [3]. Macrophages are key cellular mediators of innate immunity in OA [4] through their capacity to activate cytokines and MMPs by chondrocytes, but also by inhibiting the gene expression of anabolic genes such as COL2 and ACAN [5]. Macrophage secretome inhibits the proliferation and viability of chondrocyte stem cells, thereby modulating cartilage repair [6]. Several subpopulations, including polarized macrophages that regulate cartilage remodeling, have been identified and might be specific drug targets for OA.

Despite major advances in knowledge in disease mechanisms, OA is orphaned of structural treatment likely related to the diversity of tissues to target and the heterogeneity in clinical profile of OA patients. This leaves a place for natural products derived from herbs and plants that have biological effects and may prevent cartilage loss and joint pain. A large variety of compounds have been tested in cells or animal models [7], providing evidence of a potential use of natural products as an alternative or complementary treatment. These compounds act through different signaling pathways (NF-κB, apoptosis, MAP kinase) and target several genes such as inflammatory cytokines and MMPs. In addition, glucosamine or chondroitin possess anti-inflammatory and antioxidant properties that may inhibit the release of OA-related cytokines and slow down the progression of OA [8]. For example, glucosamine inhibits NF-κB signaling in co-culture models of synoviocytes and chondrocytes [9], thereby reducing pain and functional disability [10,11].

*Rubus idaeus*, derived from raspberry fruits, contain flavonoids and phenols [12] that inhibit inflammatory processes. Other Rubus compounds such as *Rubus fruticosus* prevent the activation of MAPK or NFkB signaling in fibroblasts, thereby preserving the type-1 collagen against UVB [13]. Blackberry extracts reverse pathological processes in mice such as manic episodes [14], whilst *Rubus idaeus* extracts (RIE) prevent hyperlipidemia [15] and kidney stone formation [16]. Moreover, polyphenolic-enriched red raspberry extract reduces collagen breakdown in bovine chondrocytes as well as the severity of arthritis in an antigen-induced arthritis rat model [17]. Given the potential protection in cartilage loss, we investigated here the effect of RIE in cartilage and the contribution of inflammatory joint cells in chondrocyte catabolism.

## 2. Materials and Methods

### 2.1. Rubus idaeus Extracts

The hydroalcoholic (ethanol 30%/water 70%) *Rubus idaeus* aerial part was extracted from a native extract ratio of 3–5/1. The extract has been standardized to contain more than 10% of polyphenols with more than 0.4% of sanguiin H6. Briefly, sample preparation corresponded to approximately 400 mg of extract of *Rubus idaeus* into a 250 mL volumetric flask, diluted to volume with methanol, followed by 10 min of sonication and at least filtering through a PTFE 0.45 µm filter. Quantification of the target compound was performed on an HPLC Agilent 1100 HPLC system equipped with a UV detector. The separation of compounds was carried out on Atlantis T3 C18 150 × 3.0 mm; 3 µm set at 32 °C. The mobile phase consisted of acetonitrile/water (50:50) + 0.1% formic acid (eluent A) and water + 0.1% formic acid (eluent B). The gradient for eluent A was as follows: 0 min, 12%; 10 min, 20%; 30 min, 43%; 40 min, 100%; 55 min, 100%. The gradient for eluent B was the difference between 10% and eluent A gradient. The total run time was 50 min, injection volume was 2 µL, and flow rate was 0.6 mL/min. UV monitoring was performed at 280 nm for sanguiin H6 compound detection. The amount of target compound was quantified by comparing the peak area of the sample with the peak area of the reference compound of known concentration. The total phenolic content was determined by using the Folin–Ciocalteu assay as previously described [18]. Colorimetry of total phenolics was performed with phosphomolybdic-phosphotungstic acid reagents. The extract powder was then diluted in PBS for cell and organ cultures just before each experiment; PBS alone was used as the control.

### 2.2. Primary Chondrocyte Culture

Primary articular chondrocytes were isolated from femoral heads, femoral condyles, and tibial plateaus of newborn C57BL/6J mice. Chondrocytes were isolated and cultured as described previously [19]. After incubation with liberase (Roche Applied Science, Penzberg, Germany) for 24 h, chondrocytes were seeded at a density of 200,000 cells/mL in 12-well plates and were expanded in Dulbecco’s Modified Eagle Medium (DMEM, Fisher Scientific, Hampton, New Hampshire, NH, USA) with 10% decomplemented Fetal Calf Serum (FCS), 2% L-glutamine, and 1% penicillin/streptomycin until confluence. To see a direct effect of RIE on chondrocytes metabolism, chondrocytes were cultured for 24 h without FCS, then confluent monolayers were stimulated with 1 ng/mL of IL1 (R&D Systems, Minneapolis, Minnesota, MO, USA) for 24 h, washed, and cultured for an additional 24 h in the presence of increasing doses of 10, 20, 40, or 100 µg/mL of RIE. To see an effect in an inflammation environment, chondrocytes were cultured for 24 h with macrophage secretome primed with RIE.

At the end of the culture, cells were lysed, and then protein and RNA were extracted for further analysis. All experiments were performed at least three times.

### 2.3. Macrophage Culture

The raw 264.7 macrophage-like cells were cultured in DMEM supplemented with 10% of decomplemented FCS, 2% L-glutamine and 1% penicillin/streptomycin. Cells were seeded at a density of 100,000 cells/well in 6-well plates and incubated for 24 h at 37 °C and 5% CO_2_. Media of each well were collected and replaced with fresh media containing RIE at 10 or 20 µg/mL. After 24 h, cells were washed and stimulated with 100 ng/mL of lipopolysaccharides (LPS) for 2 h. Then, cells were washed three times with PBS and cultured with fresh media for 24 h. Supernatants were collected and stored at −80 °C. All experiments were performed three times and cells were used at the same passage for each new experiment.

### 2.4. Cell Viability

Cell viability was determined by non-specific redox enzyme activity using CellTiter-Blue (Promega, Madison, Wisconsin, WI, USA). Macrophages were seeded in 96-well plates (100 µL, 3000 cells/well) and cultured as described above. On the last day, 20 µL of CellTiter-Blue Reagent was added to the wells and incubated for 1 hour at 37 °C protected from light. Fluorescence was recorded and values of viability of treated cells were expressed as a percentage of corresponding control cells (excitation 560 nm, emission at 590 nm). All experiments were repeated three times in duplicate.

### 2.5. Gene Expression

RNA was isolated using TRIzol Reagent (Fisher Scientific, Hampton, New Hampshire, United States) and total RNA was reverse-transcribed with a High-Capacity cDNA Reverse Transcription Kit (Fisher Scientific, Hampton, New Hampshire, NH, United States). Relative mRNA levels were evaluated by quantitative Polymerase Chain Reaction (qPCR) using a SENSIFast SYBR No-ROX kit (Bio-Technofix, Guibeville, France) and were normalized to the level of Glyceraldehyde-3-Phosphate Dehydrogenase (*Gapdh*) for chondrocytes and with TATA-Box Binding Protein (*Tbp)* and *Beta-Actin* for macrophages. Relative expression was calculated by the comparative Ct method (-∆∆Ct).

Genes expression of catabolic markers (*Mmp-3*, *Mmp-13*, *Adamts-5*), anabolic markers (*Collagen type II*, *Aggrecan*), hypertrophic markers (*Collagen type X*, *Vegf*), and inflammation (*Il1*, *Il6*, *Cox2*, *Tnf*) were evaluated. Primers sequences were as follows:

*Mmp-3*, 5′-AAAGACAGGCACTTTTGGCG-3′ and 5′-CATGTTGGATGGAAGAGATGGC-3′;

*Mmp-13*, 5′-TGATGGCACTGCTGACATCAT-3′ and 5′-TGTAGCCTTTGGAACTGCTT-3′;

*Adamts-5*, 5′-TCAGCCACCATCACAGAA-3′ and 5′-CCAGGGCACACCGAGTA-3′;

*Aggrecan*, 5′-CAGGGTTCCCAGTGTTCAGT-3′ and 5′-CTGCTCCCAGTCTCAACTCC-3′;

*Collagen type II* (*Col2a1*), 5′-CCGTCATCGAGTACCGATCA-3′ and 5′-CAGGTCAGGTCAGCCATTCA-3′;

*Collagen type X* (*Col10a1*), 5′-AAGGAGTGCCTGGACACAAT-3′ and 5′-GTCGTAATGCTGCTGCCTAT-3′;

*Vegf*, 5′-GGAGATCCTTCGAGGAGCACTT-3′ and 5′-GGCGATTTAGCAGCAGATATAAGAA-3′;

*Gapdh*, 5′-AACTTTGGCATTGTGGAAGG-3′ and 5′-ACACATTGGGGGTAGGAACA-3′;

*Il1*, 5′-CCTCACAAGCAGAGCACAAG-3′ and 5′-AAACAGTCCAGCCCATACTTTAG-3′;

*Il6*, 5′-CAAAGCCAGAGTCCTTCAGAG-3′ and 5′-GGATGGTCTTGGTCCTTAGC-3′;

*Cox2*, 5′- CCAGCACTTCACCCATCAGTT-3′ and 5′- ACCCAGGTCCTCGCTTATGA-3′;

*Tnf*, 5′-CCCATGTTGTAGCAAACCCTC-3′ and 5′-TATCTCTCAGCTCCACGCCA-3′;

*Beta-Actin*, 5′-GGCTGTATTCCCCTCCATCG-3′ and 5′-CCAGTTGGTAACAATGCCATGT-3′;

*Tbp*, 5′-CACGGACAACTGCGTTGATTT-3′ and 5′-GCTGCTAGTCTGGATTGTTCTTCA-3′.

### 2.6. Culture of Cartilage Explants

To assess the effect of RIE in the joints, we used cartilage explants obtained from femoral heads of 10-week-old C57BL/6J mice as described [20]. Femoral head explants were cultured in FCS-free DMEM for 24 h, and then stimulated for 72 h with 10 ng/mL of IL1 with or without 10 or 100 µg/mL of RIE. Explants were prepared for cryosection (n = 3 explants per condition).

### 2.7. Histology and Immunohistochemistry

Cartilage explants were fixed with 4% PFA for 24 h at 4 °C, and then decalcified with 0.5 M EDTA at room temperature for 24 h, changing the solution twice. Frozen sections (5 µm thick) of Optimal Cutting Temperature (OCT)-embedded cartilage explant were cut for histology and immunohistochemistry procedures. Cartilage breakdown was assessed on serial sections collected in 3 different zones stained with safranin-O. Sections were stained using Mayer’s hemalum for 3 min to stain the nuclei, rinsed with water three times, and then counterstained with 0.125% Fast Green for 10 min to visualize bone tissue. After being rinsed in two successive 1% acetic acid baths, they were stained in 0.5% Safranin-O 10 min, and then rinsed in 100% ethanol. To quantify the staining loss, which reveals a loss of sulfated proteoglycans, we measure the area with loss of staining, which appears in blue and the total area in red. Results are presented as the ratio of cartilage without proteoglycan staining over the total area.

Immunohistochemistry was performed on serial sections. Primary polyclonal antibodies directed against murine NITEGE neoepitope (1:100, Fisher Scientific, Hampton, NH, USA) were used to assess aggrecanase expression and localization. A DAB Peroxidase Substrate Kit (Vector Laboratories, Burlingame, CA, USA) was used for revelation and sections were counterstained with methyl green.

### 2.8. Western Blotting

The protein expression levels in the culture medium from femoral heads or protein lysate of chondrocytes were detected by Western blot. First, 10 µg of protein lysate were resolved in 4–12% SDS-PAGE under reducing conditions and transferred to a PDVF membrane. The membrane was then blocked for nonspecific binding in casein for 1 h and incubated with primary antibody MMP-3 (1:500, ab52915, Abcam, Cambridge, UK), or COX2 (1:500, sc-1745, Santa Cruz Biotechnology, Dallas, TX, USA) overnight at 4 °C. After being washed, membranes were incubated with peroxidase-conjugated anti-rabbit for 1 h at room temperature and then, developed using a blotting detection kit (Radiance Plus Revelation ECL, Azure Biosystems, Dublin, OH, USA).

### 2.9. Statistical Analysis

To determine the statistical significance, analysis was conducted by ANOVA and then by a Mann–Whitney or Kruskal–Wallis test. *p* < 0.05 was considered statistically significant. Data are expressed as mean ± SEM from at least three experiments.

## 3. Results

### 3.1. Dose-Effect on Cell Viability

The dose-effect of RIE was assessed to test the cell viability and to choose the dose for subsequent experiments. At the dose of 40 and 100 µg/mL, RIE promoted the death of primary chondrocytes after 48 h of culture. Subsequently, further experiments were conducted with the doses of 10 and 20 µg/mL that do not alter cell viability. Similar tests were conducted on macrophages. Ten and 20 µg/mL of RIE do not alter the cell viability of macrophages (106.13 ± 5.52%, 103.59 ± 9.39%, and 96.10 ± 5.24% viable cells with LPS, 10 or 20 µg/mL RIE, respectively, compared to control cells) (Appendix A).

### 3.2. Effects of Rubus idaeus in the Expression of Catabolic and Anabolic Genes

The effects of RIE were assessed by culturing primary murine chondrocytes (Figure 1). RIE alone significantly increased the expression of *Mmp-3* at 10 and 20 µg/mL (18.70 ± 12.81 and 136.96 ± 109.06-fold, respectively, *p* < 0.05) and the expression of *Mmp-13* at 20 µg/mL (8.72 ± 5.84-fold, *p* < 0.05). In contrast, the highest dose of RIE was required to decrease the expression of *Adamts-5* at 20 µg/mL compared to controls (0.62 ± 0.07-fold, *p* < 0.05) (Figure 1A). We then tested the effect on chondrocytes primed with IL1 to mimic an inflammatory context. IL1 alone induced a significant increase in the expression of catabolic markers: 255.13 ± 99.22-fold change (*p* < 0.05) for *Mmp-3*, 7.34 ± 1.56-fold change (*p* < 0.05) for *Mmp-13*. However, there was no effect of addition of RIE and IL1 compared to IL1 alone for the expression of *Mmp-3*, *Mmp-13*, and *Adamts-5* at any dose.

Investigating the effects in anabolic genes compared to controls, IL1 alone reduced the expression of *Col2a1* (0.31 ± 0.11-fold) and *Acan* (0.36 ± 0.13-fold), both *p* < 0.05. Compared to controls, RIE alone reduced *Col2a1* (0.73 ± 0.05 and 0.43 ± 0.22 -fold at 10 and 20 µg/mL, respectively, *p* < 0.05) and *Acan* levels (0.52 ± 0.10 and 0.24 ± 0.09-fold at 10 and 20 µg/mL, respectively, *p* < 0.05) (Figure 1B). However, the addition of RIE and IL1 has no significant effect compared to IL1 alone for the expression of *Col2a1* and *Acan.* Hypertrophic markers revealed no significant changes in *Col10a1* and *Vegf* expression at RIE concentration of 10 µg/mL (1.46 ± 0.49 and 0.88 ± 0.10), but a reduction at 20 µg/mL (0.40 ± 0.06 and 0.78 ± 0.10, *p* < 0.05) compared to controls (Figure 1C). *Col10a1* expression was significantly lower in IL1-stimulated chondrocytes (0.25 ± 0.07, *p* < 0.05), whilst the addition of IL1 and RIE further decreased in the two doses, without reaching significance. Lower doses of 1 and 5µg/mL provided similar results.

### 3.3. Effects of Rubus idaeus Extracts in Cartilage Loss and Proteinases in Cartilage Explants

To test the effect of RIE for cartilage degradation, we used the model of murine cartilage explants. Explants were cultured alone or stimulated by IL1 with or without two different doses of RIE. Figure 2A shows that RIE alone has no effect compared to controls. However, RIE prevented the loss of proteoglycan assessed by Safranin O at the dose of 100 µg/mL in the presence of IL1 (3.03 ± 1.86%) compared to IL1 alone (14.84 ± 3.07%), although not statistically significantly. No effect was observed at 10 µg/mL of RIE (16.21 ± 1.21% staining loss in the presence of 10 µg/mL and IL1). Moreover, addition of RIE did not change the localization and expression of NITEGE neoepitope at 100 µg/mL in explants treated with IL1 (66.05 ± 5.87% positive cells with IL1 alone compared to 37.93 ± 10.64% positive cells with IL1 and 100 µg/mL) (Figure 2B).

### 3.4. Effects of Rubus idaeus Extracts in Inflammatory Genes in Macrophages

To test the role of RIE in inflammatory conditions, macrophages were pre-treated with two different doses of RIE, and then stimulated with LPS. RIE alone did not modify the expression of *Il1, Il6*, or *Cox2*, but slightly increased the *Tnf expression* at 20 µg/mL (1.22 ± 0.09-fold change compared to controls, *p* < 0.05). Compared to control cells, LPS alone increased the expression of inflammatory genes (507.89 ± 258.99-fold for *Il1*, 5.97 ± 1.55-fold for *Il6*, 18.41 ± 2.95-fold for *Tnf*, and 7.10 ± 1.29-fold for *Cox2*, *p* < 0.05) (Figure 3). However, there was no change in the expression of any gene expression when comparing the addition of RIE and LPS to LPS alone.

### 3.5. Effects of Secretome of Macrophages Primed with RIE in Chondrocytes

We then assessed the effect of the secretome of RIE-primed macrophages in primary murine chondrocytes. Secretome of macrophages primed with 10 or 20 µg/mL of RIE alone did not affect the expression of catabolic, anabolic, or inflammatory genes of chondrocytes in basal conditions compared to controls (Figure 4A,B). Therefore, macrophages were primed with RIE and then stimulated with LPS; the supernatant was transferred to the chondrocytes cultures. Compared to the controls, the secretome of LPS-primed macrophages stimulated the expression of catabolic genes (81.02 ± 23.52-fold for *Mmp-3*, 11.31 ± 1.56-fold for *Mmp-13*, and 3.23 ± 0.42-fold for *Adamts-5*, *p* < 0.05, Figure 4A). Compared to LPS alone, chondrocytes cultured with the secretome of RIE primed-macrophages and LPS revealed a trend toward reduced expression of *Mmp-3*, *Mmp-13*, and *Adamts-5*, but this reduction did not achieve statistical significance.

Compared to controls, the expression of *Cox2* was activated upon LPS (4.61 ± 0.94-fold, *p* < 0.05). However, the secretome of RIE-primed macrophages did not significantly change the expression of *Cox2* compared to LPS alone (5.31 ± 1.70-fold and 2.60 ± 0.47-fold) (Figure 4B). Moreover, LPS significantly reduced the expression of anabolic gene *Aggrecan* (0.43 ± 0.08-fold compared to control, *p* < 0.05), but no additional effect was observed with the secretome of RIE-primed macrophages. Neither LPS nor RIE affected the expression of the *Col2a1* gene.

To assess the expression at the protein level, the expression of MMP-3 and COX was measured by Western blot in lysates of chondrocytes cultured with RIE and/or LPS (Figure 5). Compared to controls, the MMP-3 expression was similar in chondrocytes cultured with the secretome of RIE-primed macrophages at the 2 doses. However, the MMP-3 expression decreased in the chondrocytes cultured with secretome of RIE-primed macrophages and then stimulated with LPS compared to LPS alone (10.16 ± 3.43-fold with LPS to 5.93 ± 2.06-fold and 4.70 ± 0.79-fold with 10 and 20 µg/mL of RIE, respectively) (Figure 5), although not significantly. Compared to controls, COX expression was lower in chondrocytes cultured with the secretome of RIE-primed macrophages at the two doses and stimulated with LPS (6.78 ± 2.06-fold with LPS to 3.03 ± 0.69-fold and 3.43 ± 0.45-fold with 10 and 20 µg/mL of RIE) (Figure 5). Compared to the secretome of LPS-activated macrophages alone, the secretome of RIE-primed macrophages reduced the expression of *Cox2* in chondrocytes, although not significantly.

## 4. Discussion

In the present study, RIE tend to decrease cartilage loss and to reduce chondrocyte catabolism through a reduction in *Il1* and *Il6* expression by macrophages. Assessing first the direct effect of RIE in chondrocytes, we found activation in catabolic genes and an inhibition in anabolic genes that could suggest disequilibrium of the balance leading to cartilage loss. Although no direct effect has been reported in chondrocytes, this was an unexpected result as RIE has been reported to reduce deleterious cell function such as oxidative stress and NO synthesis in macrophages and skin cells [21]. Here, we found that RIE reduced *Col10* expression, indicating that the major mechanisms are chondrocytes catabolism and hypertrophy. The latter is a main mechanism of cartilage degeneration that could compensate for the disrupted metabolic activity in chondrocytes. However, compared to chondrocytes primed with *Il1*, RIE worsened both catabolic and anabolic effects, which is not what is expected for an anti-osteoarthritic drug. Cartilage loss observed in OA involves the regulation of chondrocyte function that is under the influence of interactions between cartilage matrix and chondrocytes and diffusible factors released from synovial tissues and subchondral bone [21,22,23]. To better delineate the effects at the tissue level, we tested RIE in murine cartilage explants and showed that RIE prevented the degradation of sulfated proteoglycan as revealed by Safranin O staining. These data are in line with the reduction in proteoglycan release from bovine cartilage explants in which RIE altered the collagenolytic activity via a downregulation of MMP-2 and MMP-9. This was further confirmed in A549 cancer cells [24]. These data suggest that the protection in cartilage damage is partly mediated by MMPs that preserve the breakdown of the extracellular matrix. However, subchondral bone could have contributed to the prevention of proteoglycan loss. Although no effect of RIE has been described in bone cells, RIE could have modified the release of MMP or inflammatory cytokines by bone cells or bone marrow cells that, in turn, could prevent NITEGE neoepitope expression or cartilage loss. Nevertheless, RIE alone did not protect from the catabolic nor the anabolic pathway, but RIE conversely promoted catabolic signaling, which does not make it suitable for an OA drug. Further experiments using animal models for the development of OA in mice are awaited.

Red raspberry extracts possess anti-inflammatory properties as shown by the reduction in joint lesions in antigen-induced arthritis mice [17]. Degraded matrix products promote the release of several cytokines and other inflammatory factors by the synovial cells that, in turn, induced chondrocyte catabolism [1]. Therefore, we speculated that the low-grade inflammation associated with cartilage damage could be dampened by RIE in OA. Here, there was a trend towards of a reduction in LPS-induced *Il1* and *Il6* expression by RIE that contribute to reduce the inflammatory response of the macrophage. The lack of significance may be due to the lack of statistical power. Of note, RIE failed to inhibit the expression of *Cox2* as well as *Tnf* expression as previously demonstrated in skin cells. This might be related to various inflammatory responses according to cell types. Because synovial inflammation plays an important role in cartilage breakdown, we hypothesized that macrophage secretome primed by RIE could modify the chondrocyte function. Hence, secretome of RIE-primed macrophages dose dependently decreased the expression of *Mmp-3*, *Mmp-13*, and *Adamts-5* without reaching significance. This indicates that the inhibition of expression of matrix proteases is also driven by cytokine concentration derived from the joint tissue environment. Interestingly, the impact of RIE was restricted to catabolic processes since *Col2A1* or *Aggrecan* anabolic genes were not modulated. This is contrasting to the upregulation of type 1 collagen synthesis in skin fibroblasts [25] driven by the MAPK pathway, which, again, illustrates a cell-dependent effect of RIE. However, experiments in explants showed that RIE alone prevented cartilage loss at a high dose and the secretome of RIE-primed macrophages also reduced the protein expression of MMP-3 and Cox-2. Although not identified, several factors produced upon RIE stimulation such as PGE2 or ROS could inhibit protease expression through the MAP kinase pathway [26] or an epigenetic regulation of *Cox2* as observed in fibroblasts [27]. In addition, the main effects are shown at the protein levels, suggesting that secretome could induce post translational modifications of *Cox2* that contribute to reduce chondrocyte activity.

Altogether, this in vitro study provides insights that RIE tend to reduce the content of the cartilage matrix as well as a trend towards a reduction in inflammation response in order to prevent the release of metalloproteinases. Therefore, further experiments are required to show that RIE could have an effect in cartilage breakdown. More studies are needed to investigate the effect of RIE for the prevention of osteoarthritis.

## Figures and Tables

**Figure 1 biomolecules-11-00245-f001:**
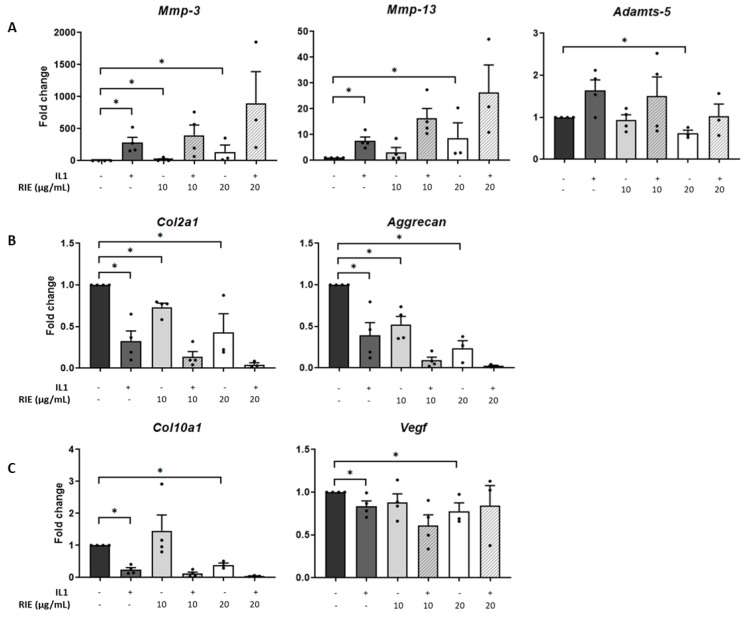
Effects of *Rubus idaeus* in catabolic, anabolic, and hypertrophic genes in primary murine chondrocytes. qPCR analysis of catabolic genes *Mmp-3*, *Mmp-13*, and *Adamts-5* (**A**), anabolic genes *Col2a1* and *Acan* (**B**), and hypertrophic genes *Col10a1* and *Vegf* (**C**) in chondrocytes stimulated with IL1 and then treated 24 h with *Rubus idaeus* extracts (RIE) (10 or 20 µg/mL). Data are mean ± SEM from four experiments. * *p* < 0.05.

**Figure 2 biomolecules-11-00245-f002:**
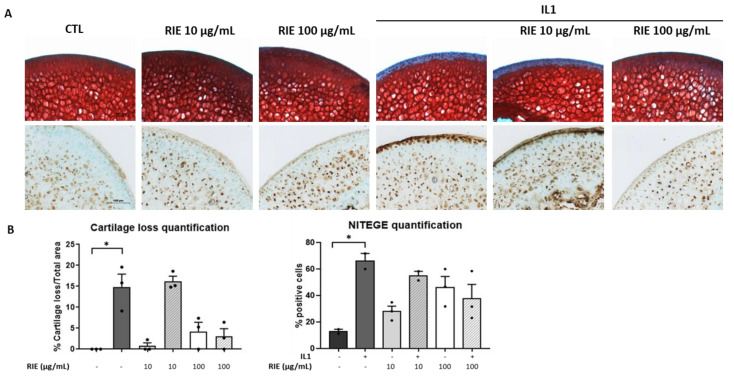
Effect of RIE in cartilage composition and NITEGE expression in murine cartilage explants. Safranin O staining, NITEGE neoepitope immunohistochemistry (**A**) and the associated quantification (**B**) of cryosections of mouse femoral head cartilage explants treated or not with IL1 and RIE (10 or 100 µg/mL) for 72 h. * *p* < 0.05.

**Figure 3 biomolecules-11-00245-f003:**
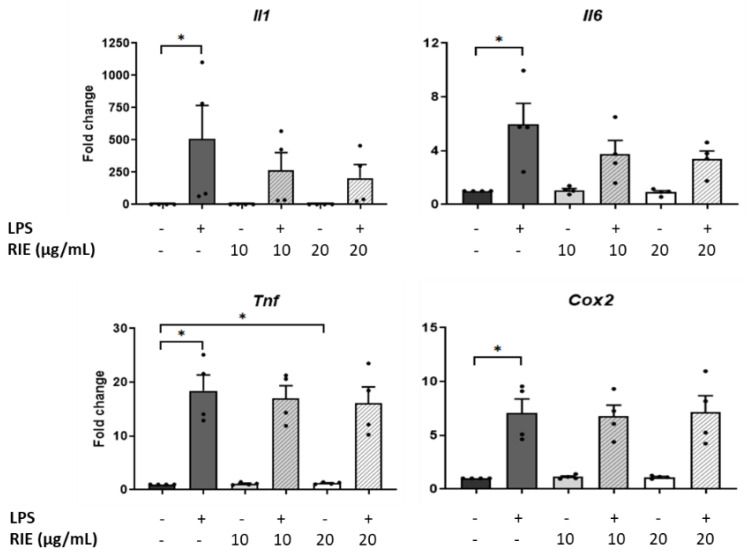
Expression of inflammatory cytokines in murine macrophages primed with *Rubus idaeus*. qPCR analysis of inflammatory genes, *Il1*, *Il6*, *Tnf*, and *Cox2* in macrophages pre-treated with RIE at two different doses (10 or 20 µg/mL) and then stimulated with LPS (100 ng/mL). Data are mean ± SEM from four experiments. * *p* < 0.05.

**Figure 4 biomolecules-11-00245-f004:**
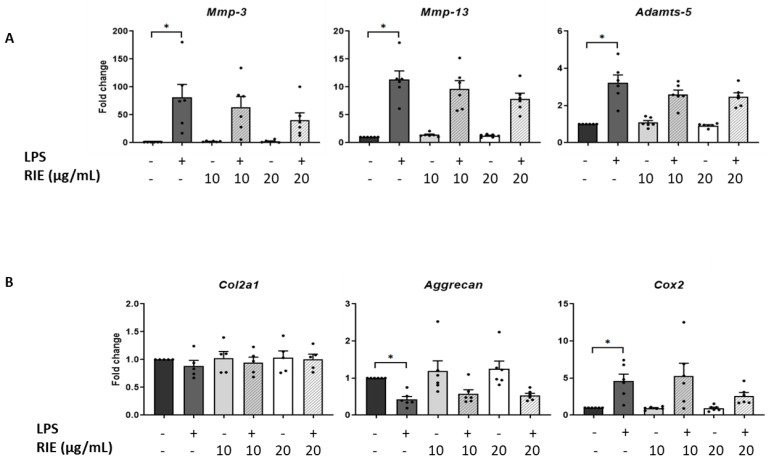
Effects in murine chondrocytes cultured with supernatant macrophage primed with *Rubus idaeus* extracts. (**A**) qPCR analysis of catabolic genes *Mmp-3*, *Mmp-13*, and *Adamts-5*, (**B**) anabolic genes *Col2a1* and *Aggrecan*, and inflammatory gene *Cox2* in chondrocytes cultured with macrophages supernatant pre-treated 24 h with RIE (10 or 20 µg/mL), then treated for 2 h with LPS (100 ng/mL), washed, and cultured for 24 h in fresh media. Data are mean ± SEM from six experiments. * *p* < 0.05.

**Figure 5 biomolecules-11-00245-f005:**
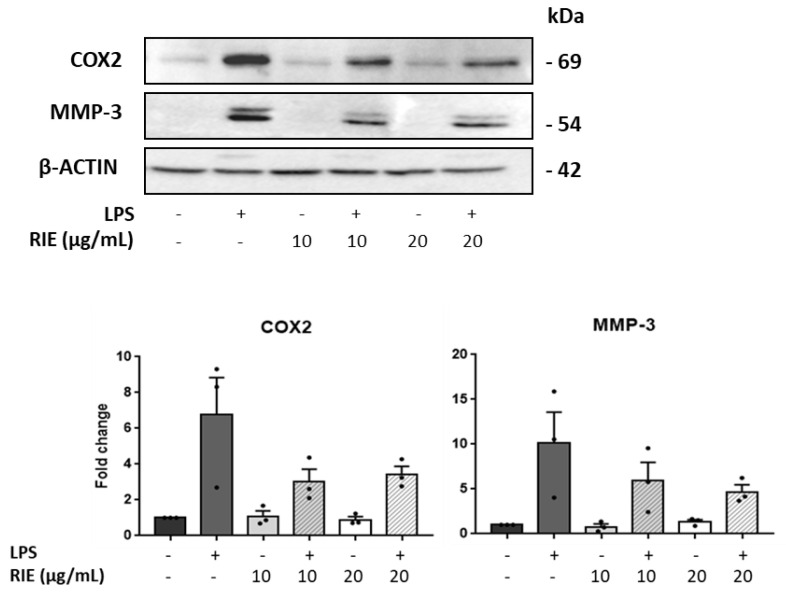
Effects of secretome of RIE-treated macrophages in *Cox2* and MMP-3 protein levels in murine chondrocytes. Western blot analysis and quantification of *Cox2* and matrix metalloproteinases MMP-3 in protein lysate of chondrocytes cultured with the supernatant of macrophages pre-treated 24 h with RIE (10 or 20 µg/mL), then stimulated for 2 h with LPS (100 ng/mL), washed, and cultured for 24 h in fresh media. Data are mean ±SEM from three experiments.

## Data Availability

Data of the study will be available upon request after agreement of the authors.

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
