# Peer review of "Effect of Rubus idaeus Extracts in Murine Chondrocytes and Explants"

_biomolecules, 2021, doi:10.3390/biom11020245_

Round 1

Reviewer 1 Report

The authors should consult a statistician for proper scientific analysis of their results. They argue that there is a significant difference between the RIE treated groups and the RIE+IL/LPS treated groups. This is not the effect they should look for, the rescue effect of RIE can only be detected by analyzing the difference between the IL or LPS treated group (mimicking injury) and the IL/LPS+RIE treated group. Looking at the results, there is no significant rescue of RIE on top of IL or LPS. Therefore, the conclusions are not supported by the results as is.

Author Response

Reviewer 1:

The authors should consult a statistician for proper scientific analysis of their results. They argue that there is a significant difference between the RIE treated groups and the RIE+IL/LPS treated groups. This is not the effect they should look for, the rescue effect of RIE can only be detected by analyzing the difference between the IL or LPS treated group (mimicking injury) and the IL/LPS+RIE treated group. Looking at the results, there is no significant rescue of RIE on top of IL or LPS. Therefore, the conclusions are not supported by the results as is.

Answer: We acknowledge the comments of the reviewer. We fully agree with his view and remodeled the manuscript accordingly. Results are now presented compared to controls in addition to IL1 / LPS and RIE in the sets of experiments. These were added in the result section and the discussion, displayed in red.

Reviewer 2 Report

The present manuscript reports on the effect of RIE on chondrocytes/cartilage degradation via reduction of inflammatory mediators.

Concerns:

  • Some more info will be helpful to understand the purification process, like final purity of the compound.
  • Macrophage culture: Could the authors explain the rational to first stimulate with RIE and then with LPS? Normally, inflammation precede the potential treatment. This approach was not done with cartilage explants.
  • Why decalcification was done on cartilage explants?
  • Western Blot: Did the authors culture also femoral heads (or this is to be intended, cartilage explants)? Please explain.
  • Figure 2: It looks that RIE alone has a negative effect on cartilage lost (min or) and NITEGE levels, compared to the control. While its effect to counteract IL-1 action is limited to the cartilage lost and at high doses of RIE. Does RIE affect cells viability at 100 ug/ml? Any analyses done on this aspect?
  • Figures 4: Although, there is a decrease in the different gene expression levels analysed, this is only true compared to LPS group, while those genes are still significant higher compared to the control. Analyses at the protein levels would help.
  • Figure 5: ELISA analyses are really needed for the quantitative evaluation of COX2 and MMP3 concentrations. Also, is the MMP3 shown in the Western Blot active or latent? Also, why check for MMP3 (stromalisine-1) when in the figure 2, the authors stained for NITEGE, a product of aggrecanase?
  • Discussion needs to be more objective, as there has not been major data supporting the last paragraph of the discussion.

Author Response

Some more info will be helpful to understand the purification process, like final purity of the compound.

Answer: fully answered

Macrophage culture: Could the authors explain the rational to first stimulate with RIE and then with LPS? Normally, inflammation precedes the potential treatment. This approach was not done with cartilage explants.

Answer: We fully agreed with the comment. This is relevant in terms of clinical management, but could be different in terms of cell biology. Our choice of this design was based on the literature and the cellular effect. The idea is that RIE with affect the gene and protein synthesis, then protein will be released by LPS.

Why decalcification was done on cartilage explants?

Answer: Yes, explants were decalcified in EDTA as described in the histology section page 8. We are used with this technique because this give better results in terms of sections. Without decalcification, the sections are fuzzy at the bone and cartilage interface.

Western Blot: Did the authors culture also femoral heads (or this is to be intended, cartilage explants)? Please explain.

Answer: the whole explants were cultured. We dissected the upper part of the femoral head. These are mainly cartilage although some subchondral bone attached to the cartilage is also included. There is no mean to dissociate the layer of subchondral bone from the cartilage beside microdissection technique, which is not available in our surrounding facilities.

Figure 2: It looks that RIE alone has a negative effect on cartilage lost (min or) and NITEGE levels, compared to the control. While its effect to counteract IL-1 action is limited to the cartilage lost and at high doses of RIE. Does RIE affect cells viability at 100 ug/ml? Any analyses done on this aspect?

Answer: RIE alone has no effect in cartilage or NITEGE expression compared to controls. We have used the concentration of 10 and 100 ug/ml for explant cultures. These concentrations are different in cell and organ cultures as quoted our references. Indeed, the concentration added in the organ cultures should be higher in order to reach the cells, likely because of the adsorption or the diffusion of the molecules by the matrix. The high dose did not affect the cell viability as shown by the NITEGE staining.

Figures 4: Although, there is a decrease in the different gene expression levels analyzed, this is only true compared to LPS group, while those genes are still significant higher compared to the control. Analyses at the protein levels would help.

Answer: We agreed. We analyzed the protein expression using immunohistochemistry. We also analyzed the expression of protein released in the supernatant of explants cultured with the secretome of RIE-primed macrophages as described in the method sections.

Figure 5: ELISA analyses are really needed for the quantitative evaluation of COX2 and MMP3 concentrations. Also, is the MMP3 shown in the Western Blot active or latent? Also, why check for MMP3 (stromalisine-1) when in the figure 2, the authors stained for NITEGE, a product of aggrecanase?

Answer: This is a very good point. Unfortunately, we do not have any supernatant left to analyzed COX2 and MMP3 by Elisa. The WB used the Ab. For the active form. We did also measure MMP3 and remove it upon request of a previous reviewer.

Discussion needs to be more objective, as there has not been major data supporting the last paragraph of the discussion.

Answer: We agreed. This was remodeled with proposed hypothesis.

Reviewer 3 Report

Bourmaud et al investigated the effect of Rubus idaeus extracts (RIE) in primary mouse chondrocytes, macrophages cell line, and mouse cartilage explants. To evaluate the effect of RIE in an inflammation environment, the authors cultured chondrocytes with the supernatant of macrophages primed with RIE. In chondrocytes, RIE significantly diminished the gene expression of Col10, but also Col2a1 and Acan, and increased the expression of catabolic genes Mmp-3 and Mmp-13. In chondrocytes cultured with the supernatant of macrophages primed with RIE, the decrease of Mmp-3, Mmp-13 and Cox2 expression was observed on gene and protein level. In cartilage explants, RIE prevented the loss of proteoglycan as well as the NITEGE, MMP-3 and MMP-13 protein expressions. RIE reduced the expression of Il1 and Il6 in macrophages, without changing Tnf and Cox2 expression. The authors conclude that RIE reduce the inflammation and thereby chondrocyte catabolism. They further suggest that RIE should be considered as a treatment option for osteoarthritis (OA).

The pathology of OA has been intensively studied and numerus compounds, also naturally occurring, were tested to provide new pharmacological options to halt or ameliorate the progression of this debilitating joint disease.

The study is well designed, and of particular importance is that the data were obtained on primary chondrocytes, to provide a better insight on what might happen in vivo. It is also applaudable that the authors performed analysis on gene and protein level as the majority of the studies measure only the expression on gene level. The experimental procedures employed are sound, the results are presented in appropriate way and the paper is well written with some minor flaws such as “fibrosblasts” in line 70 and some grammar mistakes such as line 84 “at least”.

I would only have some minor suggestions described below:

1. Please provide references to the statements in “Introduction” section describing facts about the OA, for example the first sentence in “Introduction” section.

2. Please use the abbreviations such as OA, RIE and other uniformly throughout the manuscript.

3. The abbreviations of gene names should be written in italics. Please correct.

4. Please note that the volume unit liter is abbreviated as L. In most places the authors use the correct abbreviation (L), however there are still a lot of places in Methods where l is used, for example µg/ml. Please correct.

5. Please provide the manufacturer name in the Materials and Methods for all the reagents used. Please also provide the name and manufacturer of the instruments used, for example qPCR machine.

6. Were the primers for qPCR self-designed? If not, the authors should include the reference(s).

7. The title of Figure 4 should read better: Effects in chondrocytes cultured with macrophages supernatant primed with RIE…

Author Response

Reviewer 3

Bourmaud et al investigated the effect of Rubus idaeus extracts (RIE) in primary mouse chondrocytes, macrophages cell line, and mouse cartilage explants. To evaluate the effect of RIE in an inflammation environment, the authors cultured chondrocytes with the supernatant of macrophages primed with RIE. In chondrocytes, RIE significantly diminished the gene expression of Col10, but also Col2a1 and Acan, and increased the expression of catabolic genes Mmp-3 and Mmp-13. In chondrocytes cultured with the supernatant of macrophages primed with RIE, the decrease of Mmp-3, Mmp-13 and Cox2 expression was observed on gene and protein level. In cartilage explants, RIE prevented the loss of proteoglycan as well as the NITEGE, MMP-3 and MMP-13 protein expressions. RIE reduced the expression of Il1 and Il6 in macrophages, without changing Tnf and Cox2 expression. The authors conclude that RIE reduce the inflammation and thereby chondrocyte catabolism. They further suggest that RIE should be considered as a treatment option for osteoarthritis (OA).

The pathology of OA has been intensively studied and numerus compounds, also naturally occurring, were tested to provide new pharmacological options to halt or ameliorate the progression of this debilitating joint disease.

The study is well designed, and of particular importance is that the data were obtained on primary chondrocytes, to provide a better insight on what might happen in vivo. It is also applaudable that the authors performed analysis on gene and protein level as the majority of the studies measure only the expression on gene level. The experimental procedures employed are sound, the results are presented in appropriate way and the paper is well written with some minor flaws such as “fibrosblasts” in line 70 and some grammar mistakes such as line 84 “at least”.

I would only have some minor suggestions described below:

  1. Please provide references to the statements in “Introduction” section describing facts about the OA, for example the first sentence in “Introduction” section.

Answer:  Thank you for the comment, this was indde missing. This is done now.

  1. Please use the abbreviations such as OA, RIE and other uniformly throughout the manuscript.

Answer: Thank you for noticing, this is done.

  1. The abbreviations of gene names should be written in italics. Please correct.

Answer:

  1. Please note that the volume unit liter is abbreviated as L. In most places the authors use the correct abbreviation (L), however there are still a lot of places in Methods where l is used, for example µg/ml. Please correct.

Answer: Thank you, this is done.

  1. Please provide the manufacturer name in the Materials and Methods for all the reagents used. Please also provide the name and manufacturer of the instruments used, for example qPCR machine.

Answer: Thank you, this is done.

  1. Were the primers for qPCR self-designed? If not, the authors should include the reference(s).

Answer: These were home-made designed primers.

  1. The title of Figure 4 should read better: Effects in chondrocytes cultured with macrophages supernatant primed with RIE…

Answer: Thank you again, this was done.

Round 2

Reviewer 1 Report

We appreciate the authors efforts to improve the manuscript. However, we still have a major concern regarding the beneficial/protective effects of RI in OA. As the authors show when they performed appropriate statistical analyses, there is no effect of RI in the three systems they tested: primary murine chondrocytes, murine cartilage explants, murine macrophage culture. What is more concerning is that the strongest effect observed is in primary murine chondrocytes where RI was not only not protective, but also promoted catabolic signaling, which is the complete opposite of what is desired in an OA drug. In previous versions the authors proposed an effect of RI on inflammation nation and primary chondrocytes, which was not true because the analysis was performed incorrectly. In the current version, the authors report the results properly, where RI lacks an effect on inflammatory signaling macrophages and chondrocytes. The effect of RI was limited to a modest trend in a couple of parameters/cytokines only. Therefore, the conclusion made from this study is that RI does not prevent/ameliorate inflammatory signaling in macrophages and chondrocytes, and it lacks any benefit in these in vitro models. Finally, as we stated before, the study is underpowered with high variability raising serious concerns about its rigor and reproducibility, and its ability to support any conclusions.

Author Response

Please find the answer in the attachment.

Reviewer 2 Report

Authors have answered to my previous concerns.

Author Response

Please find answer in the attachment
